A new species of freshwater crab genus Fredius Pretzmann, 1967 (Crustacea: Brachyura: Pseudothelphusidae) from a naturally isolated orographic forest enclave within the semiarid Caatinga in Ceará, northeastern Brazil

Santos Livanio C. 1 2
Tavares Marcos 3
Silva José R.F. 1
Cervini Marcelo 4
Pinheiro Allysson P. allysson.pinheiro@urca.br 2
Santana William 2 5
1 Programa de Pós-Graduação em Ecologia e Recursos Naturais, Universidade Federal do Ceará , Fortaleza , Ceará , Brazil
2 Laboratório de Crustáceos do Semiárido (LACRUSE), Universidade Regional do Cariri , Crato , Ceará , Brazil
3 Museum of Zoology, University of São Paulo , São Paulo , Brazil
4 Departamento de Ciências Biológicas, Universidade Estadual Sudoeste da Bahia , Jequié , Bahia , Brazil
5 Laboratory of Systematic Zoology (LSZ), Universidade do Sagrado Coração , Bauru , São Paulo , Brazil
Robillard Tony
Electronic publication date: 2020 Jun 29
Publication date: 2020
Volume: 8
Electronic Location ID: e9370
Received 2019 Dec 6; Accepted 2020 May 27
Copyright: ©2020 Santos et al.
Copyright year: 2020
Copyright holder: Santos et al.
License: This is an open access article distributed under the terms of the Creative Commons Attribution License, which permits unrestricted use, distribution, reproduction and adaptation in any medium and for any purpose provided that it is properly attributed. For attribution, the original author(s), title, publication source (PeerJ) and either DOI or URL of the article must be cited.
License URL: https://creativecommons.org/licenses/by/4.0/

Keywords: Fredius, Refuges, Brejos, Ibiapaba, Ipú, Amazon, Zoogeography

Funding: Fundação de Amparo à Pesquisa do Estado de São Paulo (FAPESP) 2013/01201–0 Coordenação de Aperfeiçoamento de Pessoal de Nível Superior –Brasil (CAPES) –Finance code 001 #88887.169169/2018-00 #775705/2012 Fundação Cearense de Apoio ao Desenvolvimento Científico e Tecnológico FUNCAP #BP3-0139-00166.01.00/18 Financiadora de Inovação e Pesquisa (FINEP) #1015/13 Conselho Nacional de Desenvolvimento Cientìfico e Tecnològico (CNPq) 303122/2016-1 Universidade Regional do Cariri (URCA) Universidade do Sagrado Coração (USC) Museu de Zoologia, Universidade de São Paulo (MZUSP) This work was supported by Fundação de Amparo à Pesquisa do Estado de São Paulo (FAPESP) [2013/01201–0], by Coordenação de Aperfeiçoamento de Pessoal de Nível Superior –Brasil (CAPES) –Finance code 001 (fellowship #88887.169169/2018-00 to William Santana and grant Proequipamentos #775705/2012 to Allysson Pinheiro), Fundação Cearense de Apoio ao Desenvolvimento Científico e Tecnológico (FUNCAP #BP3-0139-00166.01.00/18 to Allysson Pinheiro and fellowship to William Santana) and the Financiadora de Inovação e Pesquisa (FINEP) (#1015/13). CNPq (303122/2016-1) provided funding studies on the taxonomy of decapod crustaceans. The Universidade Regional do Cariri (URCA), Universidade do Sagrado Coração (USC) and MZUSP provided logistic support. The funders had no role in study design, data collection and analysis, decision to publish, or preparation of the manuscript.

==============================
A new species of freshwater crab, Fredius ibiapaba, is described and illustrated from a mid-altitude forested patch in Ipú (Ibiapaba plateau, Ceará, northeastern Brazil), between 635 to 782 m. The new species can be separated from its congeners by the morphology of its first gonopod: proximal half remarkably swollen, sloping abruptly downwards distally to a nearly right-angular shoulder; mesial lobe much smaller than cephalic spine; cephalic lobe moderately developed; auxiliary lobe lip, delimiting field of apical spines, protruded all the way to distal margin of auxiliary lobe. Comparative 16S rDNA sequencing used to infer the phylogenetic placement of Fredius ibiapaba n. sp. revealed that it is the sister taxon of F. reflexifrons, a species which occurs allopatrically in the Amazon and Atlantic basin’s lowlands (<100 m). Fredius ibiapaba n. sp. and F. reflexifrons are highly dependent upon humidity and most probably were once part of an ancestral population living in a wide humid territory. Shrinking humid forests during several dry periods of the Tertiary and Quaternary likely have resulted in the fragmentation of the ancestral humid area and hence of the ancestral crab population. Fredius reflexifrons evolved and spread in a lowland, humid river basin (Amazon and Atlantic basins), whilst F. ibiapaba n. sp. evolved isolated on the top of a humid plateau. The two species are now separated by a vast intervening area occupied by the semiarid Caatinga

Introduction

Cumulative evidence from many independent sources argue in favor of the mid-altitude forested patches in northeastern Brazil being remnants of a once much larger humid forest, connected to both the Amazonian and Atlantic rainforests during the moister periods (e.g., Andrade-Lima, 1982; Cartelle & Hartwig, 1996; De Vivo, 1997; Ab’Saber, 2000; Auler et al., 2004; Carnaval & Bates, 2007; Carmignotto, De Vivo & Langguth, 2012; and references therein). These humid forest refuges (Figs. 1A–1D), naturally isolated by the vast surrounding semiarid Caatinga (Figs. 1F, 1G), are indeed known to harbor many woody plant and animal species (fossil and Recent) that are also found or are closely related to species occurring allopatrically in the Amazonian and Atlantic rainforests.

Figure 1 Sítio Caranguejo, Ipú, Ceará, 04°18′50″S, 40°44′47″W, 729 m high, type locality of Fredius ibiapaba n. sp.

(A–E) Mid-altitude, naturally isolated, humid forested patch nested within the vast semiarid Caatinga domain. Note in (E) burrow (arrow) of Fredius ibiapaba n. sp. among the leaf litter. (E–F) Lowland, surrounding semiarid Caatinga forest. (E) View from above from Ipú. (F) Detail of a dry-stream channel.

Here we describe and illustrate a new species of a freshwater pseudothelphusid crab, Fredius ibiapaba n. sp., from a humid forest refuge in Ipú (Ibiapaba plateau, Ceará, northeastern Brazil), between 665 to 782 m (Figs. 1A–1D). Evidence from a phylogenetic analysis using 16S rDNA is presented for a sister taxa relationship between Fredius ibiapaba n. sp. and F. reflexifrons (Ortmann, 1897), a species occurring allopatrically in the Amazonian humid lowlands. Previous hypothesis on the phylogenetic relationships of F. reflexifrons and the possible evolutionary scenario that led to the emergence of the sister taxa Fredius ibiapaba n. sp. and F. reflexifrons are discussed.

Materials & Methods

Procedures with material examined

The specimens were collected using license permission from the Sistema de Autorização e Informação em Biodiversidade (SISBIO #29615) of the Brazilian Ministry of Environment (MMA). The studied specimens are deposited in the collections of the INPA (Instituto Nacional de Pesquisas da Amazônia, Manaus), MZUSP (Museu de Zoologia, Universidade de São Paulo, Brazil) and LACRUSE (Laboratório de Crustáceos do Semiárido). Other acronyms: SMF (Naturmuseum Senckenberg) and CCDB (Coleção de Crustáceos do Departamento de Biologia da Faculdade de Filosofia, Ciências e Letras de Ribeirão Preto). Measurements: cl (carapace length, taken along the carapace axis to the posterior median margin) and cw (carapace width, taken at the widest point), in millimeters (mm). Dates are written in the format day.month.year, with months in lower-case Roman numerals. Abbreviations are as follows: G1, G2, first and second gonopods, respectively. Mxp3, third maxilliped. The terminology used in the description of the G1 essentially follows (Rodríguez & Pereira, 1992; Rodríguez & Campos, 1998) (Fig. 2).

Figure 2 (A–B) Semi-diagrammatic view of the first male gonopod in abdominal and sternal views, respectively, with the terminology used in the descriptions.

Cl, cephalic lobe; cs, cephalic spine; fas, field of apical spines; mal, marginal lobe; mas, marginal suture; mel, mesial lobe; sab, subapical bulge.

Molecular data analysis

DNA extraction, amplification and sequencing: muscle tissue samples were obtained from the pereopods or pleon of Fredius ibiapaba n. sp., F. buritizatilis Magalhães & Mantellato in Magalhães et al., 2014, and Prionothelphusa eliasi Rodriguez, 1980. At the Laboratório de Biologia Molecular da Universidade Estadual do Sudoeste da Bahia- LBM/UESB a small region of the 16S rDNA gene was extracted with Wizard® Genomic DNA Purification Kit (Promega), amplified in a 12,5 µl final volume reaction with 2,5 mM de MgCl2 (Invitrogen), 0,05 mM de dNTP (Invitrogen), buffer 1x (Invitrogen –10xPCR Buffer: 200mM Tris-HCl (pH 8.4), 500mM KCl), 1U de taq platinum (Invitrogen) and 0,3µM of each primer. The PCR conditions were: one cycle at 94 °C, 60 s; five cycles at 94 °C, 60 s; 45 °C, 40 s and 72 °C, 60 s; and 35 cycles at 94 °C, 60 s; 51 °C, 40 s and 72 °C; 60 s; a final extension of five minutes at 72 °C was performed. The primers used were 16Sar (5′-CCGGTCTGAACTCAGATCACGT-3′) and 16Sbr (5′-CGCCTGTTTATCAAAAACAT-3′) (Palumbi et al., 1991). PCR products were purified using a polietilenoglicol (PEG) 20% and sequenced in an ABI Prism 3100 Genetic Analyzer® (Applied Biosystems) at the Departamento de Tecnologia da Universidade Estadual Paulista “Júlio de Mesquita Filho”, Jaboticabal. Sequencing reaction was performed with Big Dye v3.1 (Applied Biosystems), prepared with 4,75 µl ultrapure water, 1,5 µl BigDye 5×buffer, 0,75 µl BigDye terminator Mix, 2 µl primer (0,8 pmol) and 1 µl of Purified PCR product. Sequence conditions were: one minute at 96 °C; 35 cycles of 15 s at 96 °C; 15 s at 50 °C and 2 min at 60 °C. Both, forward and reverse sequence strands were obtained and the consensus generated by the software BioEdit 7.0.5 (Hall, 2005). The identities of the final sequences were confirmed with a BLAST (Basic Local Alignment Search Tool) on GenBank database. Additional comparative sequences were retrieved from GenBank (Table 1).

Table 1 Species of Fredius Pretzmann, 1967, Prionothelphusa Rodriguez, 1980 and Trichodactylus Latreille, 1828 used in the phylogenetic analyses, with respective sample locality, GenBank accession number and catalogue number of the voucher specimen.

Species	Locality	GenBank accession numbers	Catalogue number	
Fredius buritizatilis	Ji-Paraná, Rondônia, Brazil	JN402376	INPA 1891	
Fredius buritizatilis	Ji-Paraná, Rondônia, Brazil	JN402377	CCDB 342	
Fredius buritizatilis	Chupinguaia, Rondônia, Brazil	MN787136	LACRUSE002	
Fredius denticulatus	Serra do Navio, Amapá, Brazil	JN402372	INPA 582	
Fredius estevisi	Posto Indígena Parafuri, Roraima, Brazil	JN402379	INPA 839	
Fredius fittkaui	Aldeia Balawa-ú, Amazonas, Brazil	JN402373	INPA 1330	
Fredius platyacanthus	Comunidade Paapi-ú, Roraima, Brazil	JQ414023	INPA 841	
Fredius ibiapaba n. sp.	Sítio Caranguejo, Ipu, Ceará, Brazil	MN787135	LACRUSE001	
Fredius reflexifrons	Rio Chumucuí, Bragança, Pará, Brazil	JN402378	INPA 1512	
Fredius stenolobus	Rio Tawadu, Bolívar, Venezuela	JN402374	INPA 833	
Fredius stenolobus	Aldeia Palimi-ú, Rio Uraricoera, Roraima, Brazil	JN402375	INPA 848	
Prionothelphusa eliasi	Japurá, Vila Bittencount, Amazonas, Brazil	MN787137	LACRUSE003	
Trichodactylus dentatus	Bahia, Brazil	FM208777	SMF 32763	

Phylogenetic analyses: substitution saturation in 16S rDNA was tested using the saturation index implemented in DAMBE 5 (Xia, 2013). The sequences were grouped and edit in BioEdit and aligned using the ClustalW interface (Thompson, Higgins & Gibson, 1994). Prionothelphusa eliasi (Pseudothelphusidae) and Trichodactylus dentatus H. Milne Edwards, 1853 (Trichodactylidae) were chosen as outgroups. The best-fit model HKY + G was selected using jModeltest 2.1.7 (Darriba et al., 2012). This model was used to generate Maximum Likelihood gene trees in MEGA 6.06 (Tamura et al., 2013). Branch support values were calculated using bootstrap analyses with 1,000 replicates (Felsenstein, 1985). Only nodes with bootstrap support greater than 50 are shown on the phylogenetic tree. Nucleotide divergence estimated from pairwise distance was calculated in MEGA 6.06 with the same best-fit model (Table 2).

Table 2 Pairwise distance matrix from the portion of the mitochondrial 16S rRNA based on ∼560 bp.

		1	2	3	4	5	6	7	8	9	10	11	12	
1	Fredius ibiapaba n. sp.	–	–	–	–	–	–	–	–	–	–	–	–	
2	Fredius reflexifrons	0,04	–	–	–	–	–	–	–	–	–	–	–	
3	Fredius burutizatilis	0,10	0,07	–	–	–	–	–	–	–	–	–	–	
4	Fredius buritizatilis	0,11	0,08	0,02	–	–	–	–	–	–	–	–	–	
5	Fredius buritizatilis	0,11	0,08	0,02	0,00	–	–	–	–	–	–	–	–	
6	Fredius denticulatus	0,12	0,09	0,08	0,08	0,08	–	–	–	–	–	–	–	
7	Fredius stenolobus	0,10	0,07	0,07	0,06	0,06	0,10	–	–	–	–	–	–	
8	Fredius stenolobus	0,10	0,07	0,07	0,06	0,06	0,10	0,00	–	–	–	–	–	
9	Fredius estevisi	0,11	0,07	0,07	0,06	0,06	0,09	0,02	0,02	–	–	–	–	
10	Fredius fittkaui	0,09	0,07	0,07	0,07	0,07	0,09	0,08	0,08	0,08	–	–	–	
11	Fredius platyacanthus	0,10	0,07	0,06	0,06	0,06	0,09	0,02	0,02	0,02	0,08	–	–	
12	Prionothelphusa eliassi	0,16	0,13	0,12	0,12	0,12	0,13	0,12	0,12	0,12	0,13	0,12	–	
13	Trichodactylus dentatus	0,22	0,19	0,21	0,21	0,21	0,22	0,21	0,21	0,22	0,20	0,21	0,22	

Registration of nomenclatural act

The electronic version of this article in Portable Document Format (PDF) will represent a published work according to the International Commission on Zoological Nomenclature (ICZN), and hence the new names contained in the electronic version are effectively published under that Code from the electronic edition alone. This published work and the nomenclatural acts it contains have been registered in ZooBank, the online registration system for the ICZN. The ZooBank LSIDs (Life Science Identifiers) can be resolved and the associated information viewed through any standard web browser by appending the LSID to the prefix http://zoobank.org/. The LSID for this publication is: [urn:lsid:zoobank.org:pub:0925982D-7441-120 4256-9856-A553987956A6]. The online version of this work is archived and available from the following digital repositories: PeerJ, PubMed Central and CLOCKSS.

Results

Family Pseudothelphusidae Ortmann, 1893	
Genus Fredius Pretzmann, 1967	
Fredius ibiapaba n. sp. (Figs. 3A–3E; Figs. 4A–4C; Fig. 5A, 5C; Figs. 6A–6D; Figs. 7A–7E)	
Fredius reflexifrons –Magalhães et al., 2005: 94, fig 1 –Santos et al., 2020: 3.	

Type material. Holotype, Ceará, Ipú, Sítio Caranguejo, 04°18′50″S, 40°44′47″W, 729 m, xii.2017, male cl 36 mm, cw 53 mm (MZUSP 39710). Paratypes: Same data as holotype, male cl 34 mm, cw 48 mm (MZUSP 39169); Ceará, Ipú, Sítio Gameleira, 04°17′17″S, 40°44′44″W, 665 m, 5.i.2018, female cl 35 mm, cw 49 mm (MZUSP 39171); Ceará, Ipú, Sítio Santa Cruz, 04°19′40″S, 40°45′09″W, 782 m, 10.x.2014, male cl 32 mm, cw 48 mm (MZUSP 39167); Ceará, Ipú, Sítio Santa Cruz, 04°19′40″S, 40°45′09″W, 782 m, 23.iv.2015, female cl 31 mm, cw 44 mm (MZUSP 39168); Ceará, Ipú, Sítio Ipuçaba, 798 m, 27.xii.2017, male cl 41.2 mm, cw 62.6 mm (MZUSP 39742). Ceará, Ipú, Sítio Gameleira, quintal do Kindó, 04°17′42″S, 40°44′43″W, L.C. Cruz, J.G. Araújo, H.S. Mattos and J.E.P Araújo coll., 665 m, 01.v.2018, 3 males, cl 35.5 mm, cw 52.2 mm, cl 37.7 mm, cw 56.6 mm, cl 32.2 mm, cw 46.6 mm (LACRUSE 259). Ceará, Ipú, Sítio Santa Cruz, 04°19′40″S 40°45′09″W, L.C. Cruz coll., 782 m, 23.iv.2015, 2 males, cl 28.7 mm, cw 42.4 mm, cl 31.5 mm, cw 46.0 mm, 1 female, cl 37.7 mm, cw 55, one mm (LACRUSE 216).

Figure 3 Fredius ibiapaba n. sp., male cl 36 mm, cw 53 mm (MZUSP 39710).

(A–B) Habitus, dorso and ventral views, respectively. (C) Cephalothorax, frontal view. (D–E) Right and left chelipeds in lateral view, respectively. Scales: A–E, 10 mm.

Figure 4 (A–F) Right male first gonopod (G1) in pleonal (tilted left), lateral and mesial views from A–C and D–F, respectively.

(A–C) Fredius ibiapaba n. sp., holotype, male cl 36 mm, cw 53 mm (MZUSP 39710). (D–F) Fredius reflexifrons (Ortmann, 1897), male cl 73.8 mm, cw 53 mm (MZUSP 13178). Note in (B, C) the G1 remarkably swollen, sloping abruptly downwards anteriorly to a nearly right-angular shoulder (arrow), and in (E, F) the G1 shoulder clearly more gently sloping distally (arrow). Scales: A–F, two mm.

Non-type material. Ceará, Viçosa do Ceará, Fonte do Caranguejo, 03°33′43.2″S, 41°5′09.6″W, M. Pereira coll., 24. vi. 2004, 2 males (INPA 1382).

Comparative material. Fredius fittkaui (Bott, 1967): Guyana - Potaro-Siparuni, Rio Kuribrong, 05°22′35″N, 59°33′4″W, P. Bernardo and B. Newman coll., 28.ix.2010, male, cl 47.1 mm, cw 66.9 mm (MZUSP 24497). Fredius reflexifrons (Ortmann, 1897): Brazil - Amapá, Serra do Navio, Serra do Veado, Projeto Diversitas Neotropica, M. Tavares coll. 7.v.1994, male, cl 37 mm, cw 52 m (MZUSP 19922). Amapá, Rio Jari, montante, Cachoeira Santo Antônio, M. Jegú and J. Zuanon coll., 9-26.vi.1981, 2 males, cl 42 mm, cw 57.7 mm, and cl 53 mm, cw 73.8 mm (MZUSP 13178). Amapá, Serra do Navio/ Serra do Veado, 07.v.1994, male (INPA 583). Amapá, Laranjal, 16.i.2012, male (INPA 2125). Amazonas, Manaus, Reserva do Km 41, 02°26′56″S, 59°46′13″W, male (INPA 889). Amazonas, Manaus, Reserva Ducke, 22.ii.1986, male (INPA 368). Amazonas, Manaus, 11.vii.2001, male (INPA 850). Amazonas, Iranduba, Sítio Anaíra, 03°10′39″S, 60°07′39″W, 12.ix.1999, male (INPA 852). Pará, Santarém, Com. Santa Rosa, male (INPA 1254). Pará, Rio do Peixe Boi, 01°11′30″S, 47°18′54″W, E. Matos and A. Henriques Jr coll., 03.iii.1995, male (INPA 851). Pará, Bragança, Rio Chumucuí, S. Alves coll., 12. xi. 2004, male (INPA 1512). Peru: Rio Apiacu, Departamento Loreto, Boris Malkin coll., 15-25.iv.1966, male, cl 31 mm, cw 42.5 mm (MZUSP 6389). Fredius denticulatus (H. Milne Edwards, 1853): Brazil - Rio Amapari, Serra do Navio, AP, Projeto Diversitas Neotropica, no 151, M. Tavares coll., 30.iv.1994, C. Magalhães det. 16.ii.1996, male cl 45 mm, cw 62 mm (MZUSP 16294).

Figure 5 (A–D) Right male first gonopod (G1) in sternal and apical views from A to B and C to D, respectively.

(A, C) Fredius ibiapaba n. sp., holotype, male cl 36 mm, cw 53 mm (MZUSP 39710). (B, D) Fredius reflexifrons (Ortmann, 1897), male cl 73.8 mm, cw 53 mm (MZUSP 13178). Note in (A) and (C) the G1 apex much less tilted so that the mesial lobe is not visible in sternal view (arrow), and the subapical bulge markedly less swollen (arrow), respectively. Note the opposite in (B) and (D). Scales: A–B, two mm; C–D, one mm.

Figure 6 (A–H) Right male first gonopod (G1) in sternal, lateral, mesial, and pleonal views from A–D and E–H, respectively.

(A–D) Fredius ibiapaba n. sp., holotype, male cl 36 mm, cw 53 mm (MZUSP 39710). (E–H) Fredius reflexifrons (Ortmann, 1897), male cl 73.8 mm, cw 53 mm (MZUSP 13178). Scales: A–H, one mm.

Type locality. Sítio Caranguejo, Ipú, Ceará, 04°18′50″S, 40°44′47″W, 729 m.

Distribution. Currently known from Ipú, Ibiapaba plateau, Ceará, northeastern Brazil, in mid-altitude forests between 665 to 798 m.

Etymology. The specific epithet is a noun in apposition taken from the Tupi language word for plateau, “yby’ababa”, ibiapaba.

Diagnosis. G1 robust, proximal half remarkably swollen, sloping abruptly downwards anteriorly to a nearly right-angular shoulder (Figs. 4B, 4C); mesial lobe much smaller than cephalic spine (Figs. 4B, 4C; 5C, 5D; 7A, 7C, 7E); cephalic lobe somewhat broad, rounded apically (Fig. 4A); auxiliary lobe lip, delimiting field of apical spines, protruded all the way to distal margin of auxiliary lobe (Figs. 4A, 4B; 6B; 7D).

Description of the holotype. Carapace transversally ovate (Fig. 3A), widest at midlength (cw/cl, 1.51); dorsal surface smooth, slightly convex, regions ill-defined. Gastric pits minute, very close to each other. Cervical grooves shallow, nearly straight, poorly indicated, distal ends reaching to anterolateral margin. Front deflexed, almost straight in dorsal view, entire, marked with row of very small granules; front lower border carinate, with an almost indistinct sinus medially in frontal view; postfrontal lobules obsolete; median groove between postfrontal lobules faint (Figs. 3A, 3C). Upper orbital margin with row of very faint granules; lower margin minutely denticulate; exorbital angle marked by obtuse tooth, followed posteriorly by faint notch (Fig. 3C). Carapace anterolateral margin semicircular in outline, fringed by minute denticles; posterolateral margins almost straight, strongly convergent, smooth (Figs. 3A, 3C). Epistomial margin with minute granules; epistomial tooth broadly triangular, deflexed (Fig. 3C). Suborbital and subhepatic regions of carapace smooth; pterygostomial region densely pubescent around buccal cavity (Figs. 3B, 3C).

Figure 7 Fredius ibiapaba n. sp., paratype, male cl 41.2 mm, cw 62.6 mm (MZUSP 39742).

Scanning electron microscopy of the first right male gonopod. (A) mesial (tilted right), (B) sternal, (C) apical, (D) lateral, and (E) mesial views. Scales: A–E, one mm.

Mxp3 palp slender, long, reaching slightly beyond articulation of merus and ischium when folded. Merus markedly operculiform. Posterior half of mesial margin of merus and mesial margin of ischium with conical teeth (Fig. 3C). Exopod short, 0.28 times length of lateral margin of ischium, devoid of flagellum. Efferent branchial channel opening subcircular (Fig. 3C).

Chelipeds moderately heterochelous, right cheliped larger than left one (Figs. 3E, 3F). Major cheliped merus subtriangular in cross-section; lateral surface smooth, with irregular row of small tubercles of different sizes along dorsal surface; mesial surface smooth, slightly concave to fit lateral sides of carapace; mesial lower margin with row of conical teeth slightly increasing in size distally; lateral lower margin with row of small teeth. Carpus smooth dorsally; mesial margin with row of small, irregular teeth and strong, acute spine about midlength of margin. Palm moderately swollen, smooth on lateral and mesial surfaces, with minute granules on rounded dorsal and ventral faces. Dactylus in process of regeneration. Cutting margin of dactylus and fixed finger both with larger teeth interspersed with smaller ones. Fingers not gaping when closed, tips not crossing. Minor cheliped similar in shape.

Thoracic sternal suture 2/3 complete, distinct; sternal suture 3/4 interrupted, visible only laterally (Fig. 3B); sternal sutures 4/5 and 5/6 interrupted, ending just before reaching midline of thoracic sternum; sternal sutures 6/7 and 7/8 complete. Midline of thoracic sternum deeply incised in sternites VII and VIII.

All pleonal segments free. Lateral margins of male telson slightly concave, tip rounded (Fig. 3B).

G1 robust (Figs. 4A–4C), proximal half remarkably swollen, sloping abruptly downwards distally to a nearly right-angular shoulder (Fig. 4B, 4C). Subapical bulge moderately developed around lateral and sternal sides (Figs. 4B; 5A, 5B; 6A). Marginal suture straight (Fig. 4C). Marginal lobe truncate, projected distally beyond pleonal surface, junction with marginal lobe marked by distinct depression. Mesial lobe much smaller than cephalic spine, showing as triangular, acute spine, pointing to pleonal direction (Figs. 4A–4C, 5B; 5C; 6A–6C). Cephalic spine very strong, acuminate at tip, pointing to mesial direction (Figs. 4A–4C, 5B; 5C; 6A–6C). Cephalic lobe prominent, blunt, tip rounded, with several spinules along lateral, mesial and sternal sides (Fig. 4A, 4B; 6A, 6B). Auxiliary lobe much shorter than cephalic lobe in pleonal view, separated from it by distinct depression, their junction forming lateral channel running distally in almost straight direction before ending in inward curve subterminally (Figs. 4A; 6A). Field of apical spines large, open, flattened, elongated, ear-shaped, provided with small spinules, delimited by lateral and pleonal lips of apex (Fig. 4A, 4B; 5B; 6A, 6A).

G2 slightly longer than G1; very slender, tapering distally progressively, distal part moderately flattened, with somewhat dense, minute spinules along sternal side.

Remarks. Fredius ibiapaba n. sp. is herein assigned to the genus Fredius, whose diagnostic characters (Rodríguez, 1982; Rodríguez & Pereira, 1992) are readily recognized in the new species, namely, exopod of mxp3 short, about 0.3 times length of outer margin of ischium with G1 widest at base (Figs. 4B, 4C); marginal lobe simple, ending in an inverted cup-shaped elongation at base of field of apical spines; subapical bulge covering lateral and sternal sides; field of apical spines large, open, flattened, ear-shaped, with small scattered spinules at proximal sternal border (Figs. 4A–4C; 5B; 6A, 6B).

The new species morphologically resembles Fredius denticulatus, F. fittkaui, F. reflexifrons and F. ykaa (Magalhães, 2009), in that the gonopod cephalic spine is much more developed than the mesial lobe (see Magalhães & Rodríguez, 2002: 679, fig. 1; 683, fig. 2, respectively; Rodríguez & Campos, 1998: 766, fig. 2O, 2P) (Fig. 4A, 4C; 5B; 6A, 6C), whereas other species either have the gonopod cephalic spine little larger than the mesial lobe (F. stenolobus Rodríguez & Suárez, 1994, and F. adpressus Rodríguez & Pereira, 1992), or have it much shorter than the mesial lobe (e.g., F. buritizatilis, F. platyacanthus Rodríguez & Pereira, 1992, and F. estevi Rodriguez, 1966), or have the cephalic spine and the mesial lobe similar in size (e.g., F. granulatus Rodríguez & Campos, 1998), and F. chaffanjoni Rathbun, 1905) (see Magalhães et al., 2014 and references therein).

Fredius ibiapaba n. sp. stands apart from Fredius denticulatus, F. fittkaui, F. reflexifrons and F. ykaa in having the G1 proximal half remarkably swollen on the pleonal side, sloping abruptly downwards distally to a nearly right-angular shoulder (Figs. 4B, 4C), whereas in the latter four species the G1 shoulder is clearly more gently sloping distally (Figs. 4E, 4F).

Fredius ibiapaba n. sp. closely resembles F. reflexifrons, but the following characters derived from G1 distinguish the new species from the latter species: (1) in having the auxiliary lobe lip, delimiting the field of apical spines, protruded all the way to the distal margin of the auxiliary lobe (Figs. 4A, 4B), whereas in F. reflexifrons the lip fades away well before reaching the distal margin of the lobe (Figs. 4D, 4E); (2) the subapical bulge markedly less swollen (Figs. 5A, 5C) and the G1 apex much less tilted so that the mesial lobe is not visible in sternal view (Fig. 5A), in contrast to F. reflexifrons (Fig. 5B, 5D, respectively). Also, in F. ibiapaba n. sp. the distal margin of the cephalic lobe is blunt (Figs. 4A, 6A), whereas in F. reflexifrons it tapers progressively to a distinct narrower tip (Figs. 4D, 6D).

Fredius ibiapaba n. sp. further differs from F. ykaa in that the G1 shoulder is high and robust (Figs. 4B, 4C), whilst in F. ykaa the G1 shoulder is remarkably lower; it can be easily further differentiated from F. denticulatus in that its G1 caudal lobe lacks a field of spines spirally twisted to a transverse position (viz., Rodríguez & Campos, 1998) and from F. fittkaui in having the G1 cephalic spine straight and sharply acuminate, whereas in F. fittkaui it is curved and round tipped. Morphological differentiation between female specimens is difficult.

Discussion

Phylogenetic analysis

The mitochondrial loci 16S was successfully amplified and sequenced for Fredius buritizatilis, F. ibiapaba n. sp., and Prionothelphusa eliasi. Additional sequences used were retrieved from GenBank (Table 1). Bootstrap support values are shown on nodes of the phylogenetic tree (Fig. 8). The sister species relationships between Fredius reflexifrons and the new species is well supported by high bootstrap value. The close morphological similarity between the two species also supports such relationship.

Figure 8 Phylogeny inferred from the partial mitochondrial DNA sequence of the 16S rDNA gene.

Note the sister taxon relationship between Fredius ibiapaba n. sp. and F. reflexifrons (Ortmann, 1897).

The divergence rates between Fredius reflexifrons and F. ibiapaba n. sp. (4%) is higher than between F. estevisi x F. stenolobus, F. platyacanthus x F. stenolobus and F. platyacanthus x F. estevisi all with 2% of divergence (Table 2). Morphology and molecular data hence provide evidences for the differentiation between F. ibiapaba n. sp. and F. reflexifrons.

A survey of the pseudothelphusids described from 1840 to 2004 (Yeo et al., 2008) showed that the curve of described species is still far from being asymptotic. And indeed, new species are still being discovered either by collecting in new biomes (e.g., F. buritizatilis from a palm swamp known as “buritizal”), or by revisiting the taxonomy of widely disjunct species for testing as to their conspecific identity, such as F. ibiapaba n. sp. and F. reflexifrons.

Zoogeographical notes

Fredius currently consists of 14 species (Table 3), distributed over a vast territory, which encompass five main river basins (Rodríguez & Campos, 1998; Magalhães et al., 2014): (1) the Orinoco River basin; (2) the Essequibo-Cuyuni River basin; (3) the Amazon River basin; (4) the Madeira River basin and its tributary (Machado River); and (5) the Atlantic rivers basin, a coastal drainage of small rivers in northern South American (Guyana, Suriname and French Guiana) discharging directly into the Atlantic Ocean.

Rodríguez & Pereira (1992) performed a cladistic analysis of Fredius and suggested that F. reflexifrons and F. adpressus were sister species. The purported clade F. reflexifrons / F. adpressus was presumably supported by three putative synapomorphies: (1) [G1] mesial lobe attached to back of auricular lobe; (2) basal denticle of mesial lobe present; and (3) subapical bulge well developed.

Table 3 Geographic and altitudinal distributions for the species of Fredius Pretzmann, 1967.

Species	Country	Environment	Altitude (m)	References	
F. ykaaMagalhães, 2009	Brazil (Amazon River basin)	Lowland streams	36 to 73	Magalhães, 2009	
F. adpressusRodríguez & Pereira, 1992	Venezuela (Orinoco River basin)	Lowland streams	100	Rodríguez & Pereira, 1992	
F. beccarii Coifmann, 1939	Brazil, Guyana, Venezuela, Suriname (Essequibo-Cuyuni Rivers basin)	Streams (igarapés)	50 to 752	Rodriguez & Campos, 1998; Cumberlidge, Alvarez & Villalobos, 2014; Mora-Day, Magalhães & Souki, 2009; Magalhães et al., 2014; Zanetti, Castro & Magalhães, 2018	
F. buritizatilis Magalhães & Mantellato in Magalhães et al., 2014	Brazil (Madeira River basin)	Buritizal (palm) fields	150	Magalhães et al., 2014	
F. chaffanjoni Rathbun, 1905	Venezuela (Orinoco River basin)	River’s headwaters and mid-courses	105–300	Rodríguez & Pereira, 1992	
F. cuaoensisSuárez, 2015	Venezuela (Orinoco River basin)	Highland streams	950	Suárez, 2015	
F. denticulatus (H. Milne Edwards, 1853)	Brazil, Suriname, French Guiana (Amazon and Atlantic river basins)	Streams (igarapés) and along river margins	70 to 400	Rodríguez & Pereira, 1992; Rodriguez & Campos, 1998; Magalhães et al., 2005; Magalhães, 2009; Cumberlidge, Alvarez & Villalobos, 2014; Magalhães et al., 2014	
F. estevisi Rodríguez, 1966	Brazil, Venezuela (Amazon and Atlantic rivers basins)	River’s headwaters and streams	446 to 944	Mora-Day, Magalhães & Souki, 2009	
F. fittkaui Bott, 1967	Brazil, Venezuela, Guyana (Amazon and Atlantic rivers basins)	Streams (iIgarapés) and along river margins	151 to 500	Rodríguez & Campos, 1998 ; Magalhães & Rodríguez, 2002 ; Cumberlidge, Alvarez & Villalobos, 2014; Magalhães et al., 2014; Zanetti, Castro & Magalhães, 2018	
F. granulatusRodriguez & Campos 1998	Colombia (Amazon River basin)	Lowlands	180 to 200	Rodriguez & Campos, 1998; Cumberlidge, Alvarez & Villalobos, 2014; Cumberlidge, Alvarez & Villalobos, 2014; Zanetti, Castro & Magalhães, 2018;	
F. platyacanthusRodríguez & Pereira, 1992	Brazil, Venezuela (Atlantic rivers basin)	Streams (igarapés) and mountain areas	106 to 1229	Rodríguez & Pereira, 1992; Cumberlidge, Alvarez & Villalobos, 2014; Magalhães et al., 2014; Zanetti, Castro & Magalhães, 2018	
F. reflexifrons Ortmann, 1897	Brazil, Venezuela, Suriname, French Guaiana, Peru, Guyana (Amazon and Atlantic rivers basins)	Lowland streams	37 to 200	Magalhães & Rodríguez, 2002; Magalhães et al., 2005; Cumberlidge, Alvarez & Villalobos, 2014	
F. stenolobus Rodríguez & Suárez, 1994	Brazil, Venezuela (Orinoco River basin)	Streams in rocky areas	65 to 1020	Rodriguez & Campos, 1998; Magalhães & Pereira, 2007; Cumberlidge, Alvarez & Villalobos, 2014; Magalhães et al., 2014; Zanetti, Castro & Magalhães, 2018	
Fredius ibiapaba n. sp.	Brazil (Orographic forest enclaves)	Burrows among the leaf litter, alongside little streams and water ponds inside forest stands or directly on the humid forest floor	665 to 782	Present study	

Later, however, Rodríguez & Campos (1998) reviewed the previous data and performed a new analysis in which they decided that character 1 (mesial lobe attached to back of auricular lobe) was no longer tenable and hence was eliminated from the new analysis. They also realized that the basal denticle of the mesial lobe was indeed present in F. adpressus (character 2), but was absent in all other Fredius species. They further concluded that the subapical bulge was actually “reduced” in F. adpressus and “strongly developed” in F. granulatus, F. reflexifrons, F. fittkauii, and F. denticulatus, so that these latter two characters were also removed from the new analysis. Therefore, the putative sister taxon relationship between F. reflexifrons and F. adpressus dissolved. Rodríguez & Campos (1998) put forward, instead, the hypothesis that F. reflexifrons was sister to F. fittkauii, not to F. adpressus, based on the assumption that F. reflexifrons and F. fittkauii synapomorphically share the cephalic lobe distal margin armed with several spinules. However, as found here, this character is more widely distributed being also found in F. ibiapaba n. sp. and, therefore, cannot be used to argue for the sister taxon relationship between F. reflexifrons and F. fittkauii.

Magalhães et al. (2014) performed a distance analysis based on 16S rRNA, in which F. reflexifrons was recovered as the sister taxa to (F. fittkauii (F. denticulatus (F. granulatus (F. buritizatilis (F. platyacanthus (F. denticulatus (F. stenolobus))))))). The discovery of F. ibiapaba n. sp. revealed, however, that it is actually the sister taxa of F. reflexifrons, as shown by a comparative 16S rDNA sequencing used to infer the phylogenetic placement of Fredius ibiapaba n. sp., with F. fittkauii recovered as the sister taxa to the remaining species (Fig. 8).

The distribution range of Fredius ibiapaba n. sp. is very narrow and currently restricted to a humid enclave, a small mid-altitude forested patch in Ipú (Ceará, northeastern Brazil, Figs. 1A–1E), nested within the vast semiarid Caatinga domain (Figs. 1F, 1G). The orographic forest enclaves, such as Ipú, are typically located along the slopes of plateaus, between 600 and 1,100 m, hence high enough to receive rainfall of more than 1,200 mm year−1 of Atlantic origin (Tabarelli & Santos, 2004 and references therein). These enclaves are regionally known as “Brejos” (or “Brejos de altitude” or even “Brejos nordestinos”) (Andrade-Lima, 1982; Silva & Casteletti, 2003; Tabarelli & Santos, 2004). Fredius ibiapaba n. sp. inhabits the mid-highlands of the Ibiapaba plateau, between about 635 to 782 m, where it digs burrows among the leaf litter, alongside little streams and water ponds inside forest stands or directly on the humid forest floor (Fig. 1E). In contrast, F. reflexifrons is widely distributed in the Amazon basin’s lowlands (<100 m) from as far west as Peru (Ampyiacu River, a tributary of the Amazonas River) to as far east as the Atlantic basin (French Guiana) (Magalhães, 2003). It is found in burrows alongside the “igarapés” (streams) or digs its burrows on the humid forest floor (Magalhães & Rodríguez, 2002). Magalhães et al. (2005) misidentified the specimens from the mid-highlands of the Ibiapaba plateau with F. reflexifrons and explained its presence in Ibiapaba by a migration “…eastwards as far as Serra de Ibiapaba” during the expansion of the humid tropical forest.

Fredius ibiapaba n. sp. and F. reflexifrons are highly dependent upon humidity and our view is that they most probably were once part of an ancestral population living in a wide humid territory. The shrinking humid forests during several dry periods of the Tertiary and Quaternary (Katzer, 1933; Andrade-Lima, 1953; Bigarella, Andrade-Lima & Riehs, 1975; Ab’Saber, 1977; Bigarella & Andrade-Lima, 1982; Andrade-Lima, 1982; Clapperton, 1993; Thomas, 2000; Haffer, 2001; Haffer & Prance, 2002) likely have resulted in the fragmentation of the ancestral humid area and hence of the ancestral crab population, which was split into two sister species. Fredius reflexifrons evolved and spread in a lowland, humid river basin and is now widely distributed, whilst F. ibiapaba n. sp. evolved isolated on the top of a humid plateau (Figs. 1A–1E). The two species are now separated by a vast intervening area occupied by the semiarid Caatinga (Figs. 1F, 1G).

The expansion and shrinkage of mountain, floodplain, and gallery forests, associated to complex topography are known to have affected flora and fauna (Vanzolini, 1970; Vanzolini & Williams, 1970; Vuilleumier, 1971; Andrade-Lima, 1982; Teixeira, Nacinovic & Tavares, 1986; Haffer, 1969; Haffer, 2001; Haffer & Prance, 2002; Santos et al., 2007; Leite et al., 2016). Andrade-Lima (1982) provided a number of examples of plant species that are now confined to the Brejos, isolated from the surrounding, widely distributed Caatinga. He found two floristic components in these refuges on the top of hills, one whose species and genera have mostly originated from the southeastern flora, lies further inland in the states of Alagoas and Rio Grande do Norte; and a second one in the humid mid highlands closer to the coast, especially between Pernambuco and the border of Ceará and Piauí states (referred to as the Pernambuco Centre by Santos et al. (2007)), in which the Amazonian flora are better represented (Andrade-Lima, 1982). Santos et al. (2007) found strong bootstrap support for a close floristic relationship between the Pernambuco Centre and Amazonian localities.

It has long been known that a number of freshwater fish species inhabiting the Brejos have their closest relationships with those from the Amazonian Basin (Géry, 1969; Paiva, 1978; Weitzman & Weitzman, 1982; Ploeg, 1991; Vari, 1991; Menezes, 1996; Rosa & Groth, 2004). More recently, Pinheiro & Santana (2016) described a new species of freshwater crab genus Kingsleya Ortmann, 1897 (also a Pseudothelphusidae), from a Brejo about 750 m in Arajara district, municipality of Barbalha, Ceará. Previously to their discovery Kingsleya was known from nine species inhabiting the Amazonian lowlands (Pedraza & Tavares, 2015).

Supplemental Information

Supplemental Information 1 16s sequences for all specimens analysed

The file extension is a .mts, which is used in the software Mega version 6.06 (Tamura et al., 2013).

Click here for additional data file.

We are thankful to Célio Magalhães (Instituto Nacional de Pesquisas da Amazônia) and Rafael Lemaitre (National Museum of Natural History, Smithsonian Institution) for granting access to their respective collections. We are in debt to Waltécio de Oliveira Almeida (Universidade Regional do Cariri) for providing access to optical equipment and laboratory space and to Jessica Colavite (Universidade Estadual Paulista “Júlio de Mesquita Filho”) for the help during figure preparations. This work greatly benefited from the comments of Célio Magalhães, Tomoyuki Komai (Natural History Museum and Institute, Chiba) and an anonymous reviewer.

Additional Information and Declarations

Competing Interests

Author Contributions

Field Study Permissions

DNA Deposition

Data Availability

New Species Registration

The authors declare there are no competing interests.

Livanio C. Santos conceived and designed the experiments, performed the experiments, authored or reviewed drafts of the paper, and approved the final draft.

Marcos Tavares analyzed the data, prepared figures and/or tables, authored or reviewed drafts of the paper, and approved the final draft.

José R.F. Silva and Marcelo Cervini performed the experiments, authored or reviewed drafts of the paper, and approved the final draft.

Allysson P. Pinheiro conceived and designed the experiments, performed the experiments, prepared figures and/or tables, authored or reviewed drafts of the paper, and approved the final draft.

William Santana analyzed the data, prepared figures and/or tables, authored or reviewed drafts of the paper, and approved the final draft.

The following information was supplied relating to field study approvals (i.e., approving body and any reference numbers):

The biological material was collected using license permission from the Sistema de Autorização e Informação em Biodiversidade (SISBIO #29615) of the Brazilian Ministry of Environment (MMA).

The following information was supplied regarding the deposition of DNA sequences:

The partial mithocondrial sequences of of the 16S rDNA gene of Fredius ibiapaba (MN787135), Fredius buritizatilis (MN787136) and Prionothelphusa eliasi (MN787137) are available at GenBank.

The following information was supplied regarding data availability:

Raw data is available as a Supplemental File.

The following information was supplied regarding the registration of a newly described species:

Publication LSID: urn:lsid:zoobank.org:pub:0925982D-7441-4256-9856-A553987956A6.

Fredius ibiapaba LSID: urn:lsid:zoobank.org:act:FAE32D6B-89D2-4820-834C-A06F4D27A7F3.

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
