# Peer review of "A new species of freshwater crab genus Fredius Pretzmann, 1967 (Crustacea: Brachyura: Pseudothelphusidae) from a naturally isolated orographic forest enclave within the semiarid Caatinga in Ceará, northeastern Brazil"

_PeerJ, doi:10.7717/peerj.9370_

## Round 0.1 · original submission · Major Revisions

While all the reviewers acknowledge the quality of the data and their support to describe a new species, they also raise an number of issues that need to be addressed before accepting the manuscript for publication. The reviewers have made many interesting suggestions to ameliorate and precise the manuscript that I encourage you to follow.

·

Basic reporting

Parts of the manuscript requires careful edits, in particular, the description of the G1, which is quite complicated in the structure.
Background of the study is well explained.
Structure of the article is appropriate in general.
The establishment of the new species taxon seems to be supported by the given data.
Comments and suggestions for clarification are marked directly on the PDF main text file.

Experimental design

This article describes a new species of freshwater crab with morphological and molecular evidence. It match aims and scope of the journal.
The discovery of the new species provides some insights on the phylogeny and diversity of the genus Fredius.
The authors use the mitochondrial 16S rRNA gene to assess species delimitation, but only a single gene analysis may be insufficient for estimating phylogeny. Nevertheless, the primary purpose of this article is to describe a new species, which could be justified.
Information on DNA experiments is sufficient.
Newly obtained sequence data are not yet registered in GenBank database.
I could not open the MEGA 6 file, because I am MEGA 7 user. I do not like to spend much time to check the raw data.

Validity of the findings

The discovery of the new species is important in areas of biodiversity study.
The morphological comparison is restricted to the male gonopod 1. I would like to recommend the authors to examine other characters, in particular, female characters. Only with differentiation based on male gonopod characters, it is impossible to identify female specimens morphologically.
The discussion section needs much elaboration.

Additional comments

I strongly recommend the authors to reconsider the terminology for G1 structure. Please see attached main text remarked for details.

·

Basic reporting

This manuscript brings the description of a new species of a pseudothelphusid freshwater crab of the genus *Fredius* from a forest enclave in the state of Ceará, Brazil. The work is original and it is a straightforward taxonomic work, representing a potentially relevant contribution to the knowledge of the Neotropical freshwater crab fauna. The manuscript is well written and structured according to good taxonomic articles on the description of new taxa, and conforms to professional standards. The figures are very good and appropriate as well as the tables. As I am not a native English speaker or even sufficiently fluent in English, I cannot make a correct grammatical assessment of the wording, but it seemed to me to be a clear and technically correct text, allowing for fluent reading. I only highlighted a few minor parts where the authors could double-check the wording.

However, the manuscript has some issues that should be improved. In addition to some minor corrections made throughout the text using MS Word Correction Tools, the main issues I noticed are the following:

a) The Introduction is not sufficient to clearly demonstrate the importance of this work (the introduction of this new taxon). It should bring a little more background on the distribution of the family and/or this genus in particular. I believe the paper would be much improved if the authors comment a little bit about the overall distribution of the pseudothelphusid family or at least the genus *Fredius*, and how this new species is situated regarding this distribution (in addition to its occurrence in between the Amazon and Atlantic Forest Biomes).

b) The Introduction does not give an overview of previous work on the group in the region studied (the state of Ceará, at least). It would be interesting for potential readers to know about studies on the group that has already been done in the region or at least in the state of Ceará. At least the papers of Magalhães et al. (2005) and Pinheiro & Santana (2016) should be mentioned here.

c) Concerning the publication by Magalhães et al. (2005), despite this reviewer is the senior author of that paper, I think it deserves special attention because it dealt with material from the Ibiapaba plateau and, moreover, with material coming from the very same place of the type locality of the new species described in this manuscript. As the material examined in 2005 was determined to be *F. reflexifrons*, a question is raised about the real identity of those specimens: are they co-specific with the new species? (In that case, the determination should appear on the synonymic list of the new species.) This question should necessarily be addressed by the authors, especially considering that they are aware of this paper (it is cited in the manuscript, although in another context) and that the senior author should have examined that material during his acknowledged visit to the INPA’s collection.

d) As the molecular analysis did not include all species currently assigned to the genus *Fredius* and using only the 16S mitochondrial gene, I believe that the analysis is more a genetic distance analysis than a phylogenetic analysis of the genus itself and as such, it should be discussed. In this sense, it would be nice if the authors also compare their results with those presented by Magalhães et al. (2014) for the same genus and briefly comment on it.

e) Magalhães et al. (2005) also did a similar discussion on the zoogeographic occurrence of a *Fredius* species in the Ibiapaba plateau. Although this subject is improved in the present manuscript, I think that some mention of what was discussed by Magalhães et al. (2005) should be made here.

In addition, some other minor corrections and suggestions are also made throughout the text.

Finally, I would like to add that, although it could be considered that I might have some conflict of interest (planned collaboration with any of the authors), I conducted this review with integrity and with the idea of providing constructive criticism, and I believe that I have an appropriate level of expertise to confirm that this submission is of an acceptable scientific standard and suitable to be published in PeerJ as long as the improvements suggested above are implemented

Experimental design

The research question is well defined (description of a new taxon) and the result is relevant, representing an important contribution to improve the knowledge on the diversity and distribution of the Neotropical freshwater crab fauna. The specimens data are well presented in the “Material examined” section, but I made a few suggestions to improve it. The “Discussion” is well conducted, although some mention of what was discussed by Magalhães et al. (2005) should be made. Prior literature is adequately referenced. However, as it is a taxonomy article, it seems to me that references that cite the taxonomic authorities of the taxons mentioned in the text should also be included, although this, of course, depends on the journal’s rules.

Validity of the findings

The work is scientifically sound, with rigorous and appropriate taxonomic comparisons and references. The morphological description and the figures are relevant and of excellent quality. The evidence presented, whether molecular, morphological or zoogeographic, clearly shows that the species is new and valid, making it an original and significant research contribution. I would perhaps suggest that the authors stress a bit more that *F. ibiapaba* sp. n. and *F. reflexifrons* are pseudocriptic species differentiated only by very subtle morphological characters.

Additional comments

I made several suggestions for minor modifications and corrections using MS Word Correction Tools in order to suggest improvements to the text. Please see the attached file. However, I believe that this work lacks an essential aspect: to compare your material with that studied by Magalhães et al. (2005) and define the identity of the former. As it is material of the species considered in this manuscript as a sister species of the new taxon, a question is immediately raised: are the specimens from both studies co-specific? It is imperative that you answer it. That article cannot be ignored in this regard, otherwise, your work will be incomplete and the discussion of zoogeographic aspects would be compromised.

Reviewer 3 ·

Basic reporting

.

Experimental design

.

Validity of the findings

.

Additional comments

The manuscript contains the description of a new species of the freshwater crab genus Fredius Pretzmann, 1967, based on morphological and molecular data. The title of the manuscript is precise and reflects the content. The description of the new species is according to the International Code of Zoological Nomenclature. However, the manuscript has many issues that need to be fixed by the authors before edit it in detail.
I would, therefore, suggest accepting the manuscript with Major Revision if and only if the authors carefully take into account the following suggestions/comments. – And I would be happy to receive the revise version and help the authors to improve their manuscript.
1. From the title, please precise in which genus you are describing a new species.
2. Please remove this statement from the manuscript… and references therein…
3. We need only Fig. 1E that shows the biotope of specimens of the new species. Then please, I suggest to remove Fig. 1A-D, F-G. – This is a taxonomic work.
4. Under the Material and Methods section, precise that cw: carapace width. –This is missed from the manuscript.
5. Please remove Figure 2 from the manuscript and under the Material and Methods section, please re-write this sentence … The terminology used in the description of the G1 is referred in the figure 2. Then simply provide the reference document from where you took the terminology used for the description.
6. From where exactly you extracted DNA ? You said ‘Muscle tissue samples were obtained from the pereopods or abdomen’. Then It looks like you alternatively used pereopods and abdomen. – Why? Please provide reference.
7. Why did you decide to choose Prionothelphusa eliasi and Trichodactylus dentatus as outgroups? I also noticed that there are two distant phylogenetic taxa, then I cannot trace why... In general, the outgroup should be as close possible to the ingroup. Please select the appropriate outgroup taxon and provide justification to your choice. (Re- run the phylogenetic tree).
8. I cannot see scale bars from your figures except figure 3 and figure 7. Then, I’m not convinced that all your diagnosis characters warrant for new species recognition. For example, you said G1 robust… without scale bars from, nobody can trace this… I suggested to revise all this accordingly. This holds true for the description section. After the revision, I suggest to link each statement of the diagnostic section to the corresponding figure. - This also holds true for the description section, where you have to link each statement to the corresponding figure.- What you did, seem not enough and cannot help readers to trace your work.

---

## Round 0.2 · Minor Revisions

The new review only mentions minor issues remaining to address before formal acceptance of the study. I encourage you to examine them before submitting a revised version.

·

Basic reporting

This is the re-review. Overall, the manuscript has been sufficiently improved.

Experimental design

See previous comments.

Validity of the findings

See previous comments.

Additional comments

The manuscript has been well improved, being acceptable with minor edits, which are remarked on the main text file.

·

Basic reporting

As I mentioned in the first round of review, this manuscript is a straightforward taxonomic work, well written and structured according to good taxonomic articles on the description of new taxa, and conforms to professional standards. The authors have followed several suggestions and improved the manuscript. However, there are still some minor points that need to be considered and I have marked them with the Review Tool throughout the text.

The reply letter is very detailed and explanatory and convincingly justifies some of the points whose suggestions have not been followed. Although there are a few cases in which I do not agree with the justification but I understand that they refer to a matter of point of view. Even so, I again stressed my arguments about these issues in comments inserted throughout the manuscript.

The figures are very good and appropriate, but I made some suggestions to improve and to correct tables 1 and 3.

Experimental design

Nothing substantially different from that stated in the first review, except for the following issue. As this is a taxonomic work, I think information about synonymy is very important. This species is not being described on entirely new material but instead, they are considering the material of a population that has been previously interpreted by others according to another concept. In this situation, it is important to make clear the current interpretation of what was previously published in the relevant scientific literature. Therefore, I reinforce the recommendation to include a synonymic list and made a suggestion to that effect.

Validity of the findings

No further comment.

Additional comments

The manuscript has been greatly improved, but I still have a few minor suggestions and corrections, which I marked with the Revision Tool in the files of the main text and in tables 1 and 3. Furthermore, In their reply letter, the authors explain they were not sure about the question I asked if the specimens used in the study by Magalhães et al. (2005) and in this manuscript were conspecific, and replied that they “were not sure what is meant by the reviewer” and that they have clearly stated that *F. ibiapaba* was a new species. Of course, you did; no doubt about it. However, the question was whether, if you considered that the specimens studied by Magalhães et al. (2005) (which at least the senior author had the opportunity to examine) could be considered, according to the new data and the current new interpretation, as belonging to the new species. In this case, the identification given by Magalhães et al. (2005) should be listed as a synonym for the new species in this study. For a taxonomist, this is indeed an essential issue, as it will clearly indicate to the potential reader the taxonomic history of that taxon and the concept about the specific limits applied to it in a given time, thus avoiding future misinterpretations.

---

## Round 0.3 · accepted · Accept

Dear Colleagues,

The manuscript has been well improved and most of the suggestions made by the reviewers aiming at ameliorating it have been taken into account. I am pleased to accept it for publication.